# CoarSAS2hvec: Heterogeneous Information Network Embedding with Balanced Network Sampling

**DOI:** 10.3390/e24020276

**Published:** 2022-02-14

**Authors:** Ling Zhan, Tao Jia

**Affiliations:** College of Computer and Information Science, Southwest University, Chongqing 400715, China; zl0327@email.swu.edu.cn

**Keywords:** heterogeneous information networks, network embedding, context sampling, random walk, information entropy

## Abstract

Heterogeneous information network (HIN) embedding is an important tool for tasks such as node classification, community detection, and recommendation. It aims to find the representations of nodes that preserve the proximity between entities of different nature. A family of approaches that are widely adopted applies random walk to generate a sequence of heterogeneous contexts, from which, the embedding is learned. However, due to the multipartite graph structure of HIN, hub nodes tend to be over-represented to their context in the sampled sequence, giving rise to imbalanced samples of the network. Here, we propose a new embedding method: CoarSAS2hvec. The self-avoiding short sequence sampling with the HIN coarsening procedure (CoarSAS) is utilized to better collect the rich information in HIN. An optimized loss function is used to improve the performance of the HIN structure embedding. CoarSAS2hvec outperforms nine other methods in node classification and community detection on four real-world data sets. Using entropy as a measure of the amount of information, we confirm that CoarSAS catches richer information of the network compared with that through other methods. Hence, the traditional loss function applied to samples by CoarSAS can also yield improved results. Our work addresses a limitation of the random-walk-based HIN embedding that has not been emphasized before, which can shed light on a range of problems in HIN analyses.

## 1. Introduction

Network embedding plays a crucial role in mining network data. It aims to represent the proximity between nodes by low-dimensional vectors, which can be achieved by different approaches, such as adjacency matrix factorization [1,2], inferring the spreading sequence [3,4], learning the evolution of node status [5,6], and more [7]. A family of approaches is based on learning the network representation from samples of the network. Usually, the sampling is carried out by a random walker on the network, which generates sequences of nodes visited. Analogous to sentences composed of words, these node sequences are processed by methods applied in natural language processing (NLP) [8,9]. For example, the skip-gram with negative sampling model (SGNS) [10] is often used to sample the context around the center node in a sliding window of fixed size and to encode the sampled center–context node pairs to low dimensional vectors.

Related studies are quickly extended from homogeneous networks to heterogeneous information networks (HIN) [11,12,13], which refer to networks composed of diverse node types and/or relationships between nodes [14]. Inspired by methods in homogeneous networks, random-walk-based HIN embedding methods have been proposed [15,16]. The direction of random walk between different types of nodes can be further elaborated, forming a meta-path, to obtain node sequences with heterogeneous information [17]. Despite the emergence of new frameworks, such as the graph neural network (GNN) [5,18,19], random-walk-based methods are still widely used to sample the context information of network structure [6,20,21].

The network representation approach relies on the quality of samples taken, which is supposed to be balanced, and hence accurately reflects the proximity between nodes. However, nodes with high centrality, such as the hub node, are likely to appear more frequently in the trajectory of the random walker. Therefore, the traditional framework by the random walk and the sliding window is likely to generate less balanced network samples, which, in turn, negatively affects the embedding results. First, if the target node is a node with high centrality, it may appear repeatedly in the sliding window, yielding self-pairs that start and end on the same node (node 1 in Figure 1). The self-pairs will introduce redundant information and semantic inaccuracies [22]. Second, other nodes may be associated with the high centrality node too often. For example, node 2 and node 3 in Figure 1 are both neighbors of node 1. However, because node 3’s degree is much higher, it would form more node pairs with node 1. Therefore, the learning algorithm is likely to assume that node 1 is closer to node 3 than node 2, although their topological separations are the same. These two issues are most prominent in HIN, whose multipartite graph structure is sustained by connections between different types of nodes. Such a structure effectively generates a lot of stars, in which a hub node connects multiple low-degree nodes, giving rise to a high abundance of hub nodes in a fixed sliding window.

The imbalanced sampling cannot be solved by a simple fix. Intuitively, one may consider using a dynamic window size that skips self-pairs. However, this cannot fix the issue of the high centrality nodes being over-represented in node pair samples. Alternatively, one may also consider removing the high centrality nodes in the node sequence to reduce its appearance or omitting the repeated nodes in a node sequence to avoid self-pairs. However, we also need to consider that a high degree node is naturally supposed to be sampled more often than a low degree node, as we need to gauge more distance measurements from the high degree node to its neighbors. Hence, we need a careful design that preserves the topological information. Arbitrarily removing nodes will lead to inaccurate graph sampling.

In this paper, we propose a novel sampling strategy that adequately addresses the issue of imbalanced sampling. Instead of using a sliding window on a long node sequence, we generate multiple short sequences for each center node. The number of times that a center node is sampled depends on its degree, allowing the sampling to capture sufficient information. We apply a self-avoiding sampling strategy to avoid self-pairs. We also utilize a network-coarsening step to reduce the over-representation of certain nodes in context. This sampling strategy (CoarSAS) allows us to collect the rich structure and context information more accurately for each node. We further optimize the loss function, allowing us to reach a new embedding method: CoarSAS2hvec. We consider nine embedding methods as the baseline, including two GNN-based and seven structure-preserving state-of-the-arts. CoarSAS2hvec outperforms the baseline methods in node classification [23] and community detection [24] using four real-world data sets. The ablation study demonstrates that the samples collected by CoarSAS contain a higher information entropy than other methods, hence capturing more diverse information of HIN. This adequately explains why samples by CoarSAS could give good enough results with a traditional loss function. We also find that the proposed loss function does not work equally well on samples collected by baselines. The improvement by the new loss function can only be feasible when the quality of the sample is properly controlled. The original code of CoarSAS2hvec is available at https://github.com/lzhan94swu/coarsas2hvec (accessed on 13 October 2021).for future reference and reproducibility.

## 2. Related Work

As our work is most relevant to embedding methods that preserve the proximity of nodes, we mainly focus on related works in this direction, although there are numbers of other different approaches [13]. We categorize heterogeneous network representation learning methods into two categories according to the information each method aims to preserve.

### 2.1. 1st/2nd-Order Proximity-Based Network Embedding

One essential problem in network embedding is to determine the separation of nodes. Intuitively, one can directly measure the direct neighbors (1st-order) and neighbors of the direct neighbors (2nd-order) as a collection of close enough nodes. LINE [25] has been proposed to preserve this kind of 1st- and 2nd-order proximity in homogeneous networks. The idea is soon applied in HIN. To preserve the edge-type-based 1st-order proximity between nodes, some methods build bipartite networks decomposed from the original HIN according to the edge types for training [11,26,27,28]. For example, PME [27] projects nodes into separate Euclidean embedding spaces of different semantics according to the edge types for edge prediction. To preserve the semantic association connected by meta-paths, another kind of methods construct semantic-based training networks by sampling the specified relationships or meta-paths as the intrinsic connections between nodes [18,29]. Methods to preserve the 2nd-order based proximities (neighborhood similarities) between nodes [21,30] have been recently proposed. NSHE [30] aims to constrain the type of second-order proximity between nodes by sampling neighbors for each node following the meta-schema (also known as network schema [31], a meta template for a heterogeneous network) of the HIN.

### 2.2. Random-Walk-Based Network Embedding

The use of 1st- and 2nd-order proximity gives a fixed feature or parameter to the algorithm, making it less flexible in choosing other proximity features. Another category is to sample the network by a random walker, forming a sequence of nodes that the walker successively visits. Similar to the methods in NLP, a center node is chosen, and a window is used to find context nodes around the center node along the collected sequence. All context nodes that fall into the window are considered close to the center nodes, hence lifting the constraints on a particular order of proximity. Inspired by the pioneering Deepwalk [8] and node2vec [9] in network embedding, several random-walk-based methods have been proposed. Depending on the need for particular meta-paths, they can be roughly divided into two groups. The meta-path-based family [15,16,22,32] requires artificially specified meta-paths to guide the formation of sequences. For example, metapath2vec [15] was the first work that uses a pre-determined meta-path to guide the random walk, which yields a node sequence with specified semantics. The meta-path-free approaches [33,34,35] do not constrain the meta-path of the walker. Instead, they choose the switch between different types of nodes with a specific purpose, such as balancing the types of the obtained target–context node pairs [34] or using the loss function that takes different types of paths into consideration [35]. Specifically, HIN2vec [22] learns node embeddings by predicting whether there is a certain meta-path instance between the node pairs sampled by random walk.

## 3. Methods

Our method contains two components: HIN structure sampling and HIN structure embedding. We adopted a neighbor-type-guided random walk mechanism and introduced a self-avoiding short sequence sampling approach, which avoids self-pairs and balances the appearance of center nodes. Furthermore, we added a network coarsening step to balance the appearance of the context node. Finally, we applied a modified SGNS loss to embed the sampled structure into low dimensional vectors.

### 3.1. Preliminaries

The notations in the remaining sections and their descriptions are shown in Table 1.

*Heterogeneous Information Network.* An information network is a directed graph G=(V,E) with a node type mapping ϕ:V→T and an edge type mapping ψ:E→R. Particularly, when the number of node types |T|>1 or the number of edge types |R|>1, the network is called a heterogeneous information network.

*Heterogeneous Information Network Embedding.* Given an information network G, network embedding is to learn a function $F:V\to R^{d}$ that represents each node v∈V as a vector v in a *d*-dimensional space $R^{d}$, d≪|V| that is able to preserve the structural characteristics captured by the context set $C_{v}$ of the node *v* sampled from the network. Particularly, when G is a heterogeneous information network, the corresponding process is defined as heterogeneous information network embedding.

### 3.2. HIN Structure Sampling

#### 3.2.1. Random Walk on HIN

At each step *i* of a walk, we first randomly chose the type $t_{c}$ of the next node according to the current node $v_{i}$ as
(1)$$P\left(t_{c}|v_{i}\right)=\left\{\begin{array}{ccc} \frac{1}{|T_{N(v_{i})}|} & , & t_{c}\in T_{N(v_{i})}\\ 0 & , & t_{c}\notin T_{N(v_{i})}. \end{array}\right.$$

The $T_{N(v_{i})}=\left\{\phi \left(v\right),v\in N\left(v_{i}\right)\right\}$ in Equation (Equation 1) is defined as the set of node types in neighbors of node $v_{i}$, where $N(v_{i})$ is the neighboring node set of $v_{i}$. Once $t_{c}$ is known, we randomly selected a node $v_{i+1}$ from all type $t_{c}$ nodes adjacent to $v_{i}$ as
(2)$$P\left(v_{i+1}|v_{i},t_{c}\right)\left\{\begin{array}{ccc} \frac{1}{|N_{t_{c}}\left(v_{i}\right)|} & , & \left(v_{i+1},v_{i}\right)\in E,\phi \left(v_{i+1}\right)=t_{c}\\ 0 & , & \left(v_{i+1},v_{i}\right)\notin E,\phi \left(v_{i+1}\right)=t_{c}\\ 0 & , & \left(v_{i+1},v_{i}\right)\in E,\phi \left(v_{i+1}\right)\neq t_{c} \end{array}\right.,$$
where $N_{t_{c}}\left(v_{i}\right)$ denotes the neighbor node set of type $t_{c}$ adjacent to node $v_{i}$. As different node types contain different numbers of nodes, the type selection in the first step avoids the type imbalance in the vanilla random walk [8] on HIN. Meanwhile, such random walk avoids the strict restriction on the neighbor nodes in meta-path-based random walks [15,32] and samples diversified local structures.

#### 3.2.2. Self-Avoiding Short Sequence Sampling (SAS)

For each round of a random walk starting at node $v_{0}$, we set node $v_{0}$ as the center node, and the subsequent nodes visited as context nodes, directly generating node pairs as $\left(v_{0},v_{1}\right)$, $\left(v_{0},v_{2}\right)$, …. To avoid self-pairs, we omitted pairs when the center node and context node are the same ($v_{i}=v_{0}$). The random walk stops when *l* node pairs associated with $v_{0}$ are collected.

To adequately gather the local information, we made multiple runs of a random walk on each node. As the neighborhood information is characterized by a node’s degree, a ’one-size fits all’ sampling for each node would be sub-optimal to well preserve the structure of networks [36]. Therefore, we took the number of runs on node *v*, or, equivalently, the number of samplings $q_{v}$ proportional to its degree as
(3)$$q_{v}=q\left(G\right)\cdot \frac{deg(v|G)}{2\cdot |E|},$$
where q(G) is the total sampling times on G, deg(v|G) is the degree of node *v* in network G, and |V| and 2·|E| reflect the total nodes and edges number of G, respectively.

To summarize, for a HIN G, we went through all nodes individually. For each node *v*, we set it as the starting node and ran the random walk $q_{v}$ times. For each run, we collected *l* node pairs in which node *v* cannot simultaneously appear on both sides of a pair. SAS reserves the heterogeneity of the degree distribution and retains more structure information compared to simply removing duplicate nodes in a random walk [22] or deleting self-pairs in the sliding window.

#### 3.2.3. HIN Coarsening with SAS (CoarSAS)

The SAS controls the appearance of the center nodes. However, the appearance of the context nodes may still be biased by their degree. Some hub nodes would be over-represented in the sampled node pairs as context nodes. As an example, when node 1 in Figure 2 is set as the center node, node 3 would be paired more often than node 2, giving rise to a bias in the distance measure. Likewise, nodes 4, 5, and 6 would be paired infrequently due to the hub node 3, which dilutes the second-order proximity. To overcome this problem, we applied a coarsening step to remove some over-represented context nodes.

Denote $T^{R}$ by a set of node types. For all node pairs sampled by SAS on $G_{i}$, we focused on pairs whose context node falls into node type $T^{R}$. We ranked the appearance of these context nodes, obtaining a set of the top-k nodes with the largest appearance, denoted by *R*. A hyper-parameter *a* was used to determine the percentage of cutoff for the top-k nodes. We then constructed a coarsened graph $G_{i+1}$ in which nodes in *R* are removed and the neighbors of each removed node are randomly rewired. In particular, for each node *v* in *R*, we found its neighbors N(v). For every node in N(v), we connected this node with another random node in N(v) to approximatively preserve local communities. The procedure is illustrated in Figure 2.

The above procedure introduces a hyper-parameter $T^{R}$. A default choice is to set $T^{R}=T$. However, we found that the overall performance can be further improved by tuning the node types in $T^{R}$ (i.e., $T^{R}\subset T$). More details are discussed in the ablation studies.

The coarsened network $G_{i+1}$ was sampled with the same method mentioned above. Note that the purpose of constructing $G_{i+1}$ is to strengthen the high order connection and to balance the appearance of context nodes. Nodes in *R* should not be excluded as being center nodes. Therefore, for each node v∈R, we set it as the starting node and ran the SAS on the original graph $G_{0}=G$. The heterogeneity in the sampling time for each starting node is kept as
(4)$$q_{v}=q\left(G_{i}\right)\cdot \frac{deg(v|G)}{2\cdot |E|},$$
where the degrees of nodes are counted in the original network G. To balance the contribution between the original network and the coarsened network, we adjusted the average number of sampling $q(G_{i+1})$ as
(5)$$q\left(G_{i+1}\right)=q\left(G_{i}\right)\cdot p,$$
where the hyper-parameter *p* controls the increase or decrease rate.

The network can be coarsened and sampled multiple times by the procedure described above. We set another parameter *b* as the total number of network-coarsening steps applied. For higher-order coarsened networks, the removed nodes were still sampled in the original network G as center nodes, which avoids damage to the original network structure caused by coarsening. The pseudo-code for the HIN CoarSAS procedure is illustrated in Algorithm 1.
**Algorithm 1** CoarSAS**Require:**  original HIN G=(V,E), coarsened HIN $G_{i}=\left(V_{i},E_{i}\right)$  removed node set *R*, context samples range *l*  center sampling times $q_{v}$, center node *v***Output:** center-context set $S(G_{i})$1:**for**v∈V**do**2:    **while** $q_{v}>0$ **do**3:        $S_{v}=\emptyset$4:        **if** v∈R **then**5:           $S_{v}=$ SAS(G,v,l)6:        **else**7:           $S_{v}=$ SAS$\left(G_{i},v,l\right)$8:        **end if**9:        Add $S_{v}$ to $S(G_{i})$10:        $q_{v}-1$11:    **end while**12:**end for**13:**return**$S(G_{i})$

### 3.3. HIN Structure Embedding

The CoarSAS procedure introduced above provides a collection of node pairs characterizing the co-occurrence between the center node and its context node. In this procedure, we will learn the embedding of nodes from the pairwise relationship. For simplicity, we followed the idea of SGNS [10,37] model by mapping each center node into a low-dimensional vector $v\in R^{d}$ as the output embedding, and mapping each context node into $c\in R^{d}$ as self-supervised label embedding. We trained the model to learn the embedding of the center nodes by maximizing the conditional probability of the context nodes given the center nodes. The probability of $c\in C_{v}$ given v is defined by the heterogeneous softmax function [15]:(6)$$P\left(c|v\right)=\frac{\exp\left(c^{⊤}\cdot v\right)}{\sum_{k=1}^{|V^{\phi (v)}|}\exp\left({c_{k}^{\prime }}^{⊤}\cdot v\right)},$$
where $V^{\phi (v)}$ denotes the node set of type ϕ(v), and $c_{k}^{\prime }$ denotes the *k*-th negative context nodes.

To further make use of the information from the association of different node types in the center–context pairs, we designed a pair-type embedding matrix $W\in R^{|T|\times T\times d}$. The probability in Equation (Equation 6) was modified by adding the matrix W as the relation type indicator, giving rise to a new probability function:(7)$$P\left(c|v\right)=\frac{\exp\left({\left(W_{\phi (c),\phi (v)}⊙c\right)}^{⊤}\cdot v\right)}{\sum_{k=1}^{|V^{\phi (v)}|}\exp\left({\left(W_{\phi \left(c_{k}^{\prime }\right),\phi \left(v\right)}⊙c_{k}^{\prime }\right)}^{⊤}\cdot v\right)},$$
where the ⊙ operation denotes the element-wise multiplication between vectors.

The objective function is to maximize the above probability for each center–context node pair, which is equivalent to minimizing the negative log-likelihood function $L=\sum_{v,C_{v}\in S\left(G\right)}-logP\left(C_{v}|v\right)$. By utilizing the heterogeneous negative sampling [15], the objective function was approximated as
(8)$$\begin{array}{cc} L= & \sum_{v,C_{v}\in S\left(G\right)}\sum_{c\in C_{v}}\{-\log\sigma \left({\left(W_{\phi (c),\phi (v)}⊙c\right)}^{⊤}\cdot v\right)\\ & -\sum_{k=1}^{neg}\log\sigma \left({\left(W_{\phi \left(c_{k}\right),\phi \left(v\right)}⊙c_{k}^{\prime }\right)}^{⊤}\cdot v\right)\},c_{k}^{\prime }∼P_{V^{\phi (v)}}, \end{array}$$
where neg is the number of required negative samples, and $P_{V^{\phi (v)}}$ is the distribution of nodes in $V^{\phi (v)}$. We adopted the Adam algorithm [38] to minimize the objective in Equation (Equation 8). The pseudo-code for CoarSAS2hvec is illustrated in Algorithm 2.
**Algorithm 2** CoarSAS2hvec**Require:**  an HIN G=(V,E)  coarsen round *b*, coarsen rate *a*  total sampling times q(G), negative numbers neg  context samples range *l*, attenuation ratio *p***Output:** embedding v for v∈V1:Initialize all the parameters θ={v,c,W}, S=∅2:Calculate $q_{v}$ by q(G) according to Equation (Equation 3)3:i = 0, $G_{i}=G$4:**while**i<b**do**5:    **for** v∈V **do**6:        $S(G_{i})$ = CoarSAS$\left(G,G_{i},q_{v},l\right)$7:        Add $S(G_{i})$ to S8:    **end for**9:    $G_{i+1}$ = Coarsen$\left(G_{i},a\right)$10:    $q_{v}=q_{v}\times p,i=i+1$11:**end while**12:$S(G_{b})$ = CoarSAS$\left(G,G_{b},q_{v},l\right)$13:Add $S(G_{b})$ to S14:**while** not converge **do**15:    Calculate *L* by S and neg according to Equation (Equation 8)16:    Update θ by $\frac{\partial L}{\partial \theta }$17:**end while**18:**return**v for v∈V

### 3.4. Complexity Analysis

The time cost of the sampling depends on both the SAS and network coarsening. The complexity of the SAS is O(|V|×b×l×q) because it relies on random walks that begin with each node in the network. The complexity of the network coarsening is O(|V|log|V|) because this procedure needs to count and rank the appearance of context nodes. The total complexity of the network structure sampling process is O(|V|×b×l×q+|V|log|V|) because SAS and network coarsening are two independent procedures. Since b,l,q are hyper-parameters and b×l×q≪|V|, the total time complexity of the sampling process is O(|V|log|V|). It is noteworthy that node sampling could be easily paralleled by multi-processing, which can sufficiently increase the actual speed.

## 4. Results

### 4.1. Data Description

We utilize four real-world heterogeneous network data sets of different scales from two different fields as benchmarks for evaluation, which are also widely used in other studies. *ACM* [39] is a citation network containing 4019 labeled papers divided into three classes, in which, the labels are determined by the domains of conferences. *DBLP* [40] is also a bibliographic network containing 4057 labeled authors divided into four classes, and the labels are determined by their research domains. *Aminer* [41] is a large academic network containing 164,012 labeled authors divided into 10 classes determined by their research interests. *Freebase* [42] is a knowledge graph containing 2386 books divided into seven classes according to their contents. We extract a subgraph with 10 different edge types. The statistics of the data sets are introduced in Table 2.

### 4.2. Baselines

The semi-supervised GNN-based approaches, such as GAT [43] and HAN [18], are beyond the scope of this paper. We consider nine baselines in network embedding that can be roughly divided into two categories. *1st/2nd-order proximity-based methods*: LINE [25], DGI [44], HeGAN [28], NSHE [30]. *Random-walk-based methods*: Deepwalk [8], HIN2vec [22], metapath2vec [15], HeteSpaceyWalk [32], BHIN2vec [34].

LINE [25] models and preserves the 1st/2nd proximity in networks;DGI [44] utilizes GNN to maximize the mutual information between the local structure of each node and the global structure of the network to train the GNN and learn the embeddings;HeGAN [28] adopts adversarial learning to distinguish different types of edges between real nodes and generated virtual nodes to learn the network embeddings;NSHE [30] samples neighbors of nodes according to the meta-schema of a network and utilizes the GCN to preserve the meta-schema-based 2nd-order proximity between nodes;Deepwalk [8] samples truncated random walks for nodes and learns their embeddings by applying the SGNS model;HIN2vec [22] samples node-pairs by random walks and learns embeddings by estimating the possible meta-paths between the node-pairs;Metapath2vec [15] introduces a specific meta-path-guided random walk to encode semantic information in the node embeddings and node-type-based negative sampling for the SGNS model;HeteSpaceyWalk [32] extends meta-path-guided random walks by spacing out and skipping the trivial transitions in between the meta-path, meta-schema, and meta-graph;BHIN2vec [34] regards the skip-gram-based methods as a multi-task learning process and balances different node type pairs by updating a type-pair table so as to bias the walking process.

LINE, DGI, and Deepwalk are designed for homogeneous networks, whereas other approaches are designed for HIN. Due to the fact that DGI maximizes the mutual information between local and global structures by convolution on adjacency relations, we categorize it into the first-order proximity-based method. DGI and NSHE are GNN-based state-of-the-art methods. Since the proposed method only uses self-supervised context information for network representation learning, we only compare the network representation learning algorithm based on self-supervised learning. Moreover, to independently explore the effect of the loss function in Equation (Equation 8) and the effectiveness of our proposed sampling strategy, we replace the loss function of *CoarSAS2hvec* with that of skip-gram++ model and record the results as *CoarSAS2vec*.

### 4.3. Reproducibility

#### 4.3.1. Hyper-Parameters

The codes utilized in the experiments are downloaded from the pages of the original authors. We refer to previous works [15,20,34,35] for details on setting the hyper-parameters of the baselines. For the baselines developed for homogeneous networks, we neglect node types and treat each HIN as a homogeneous network. For the common parameters, we set the training epochs iter=50, and set embedding dimension d=128. We set the embedding dimension d=128 following the most related works [15,34,35] and set the rest of the hyper-parameters following [20] because it provides a uniform parameter setting for methods of different categories.

To be more specific, for all of the competitors, we set the number of training epochs iter=50. For the random-walk-based methods, we set the walk time per node t=20 and walk length l=10, and fix the sliding window size w=5. We utilize the skip-gram++ model [15] as the uniform embedding encoder for their sampled node sequences for fairness.

For those methods requiring negative samples, we set the sample size neg=5. For those needing pre-trained embeddings or node features (HeGAN, DGI, NSHE), we utilize the embedding results of Deepwalk as their input. The meta-paths and meta-graphs utilized for metapath2vec and hetespaceywalk are summarized in Table 3, where [·||·] denotes the merging operation.

BHIN2vec and NSHE require identical edge types for nodes of the same types, and some of the paper nodes in Aminer do not have connections with “type” nodes, so we run these two methods on a subnetwork only composed of “author”, “paper”, and “conference” nodes. For the same reason, we sample a subnetwork of Freebase for these methods.

DGI and NSHE are GNN-based methods, requiring an adjacent matrix of the networks as their input. Since Aminer is too large to fit into GPU memory, we utilize its sparse form as the input of these methods.

We set the remaining parameters of each method to the default values as provided in the source code. We summarize the parameter settings of CoarSAS2hvec on each data set in Table 4.

#### 4.3.2. Environment

We run all the non-GNN methods on a single Linux server with an Intel(R) Core(R) i9-7900X CPU @3.3GHz, 96G RAM, and two NVIDIA Turing RTX2080Ti-11GB GPUs. We overclock the CPU to 4.5GHz to speed up the calculation.

For small data sets, including ACM and DBLP, we run all the GNN-based methods on the above server. For large data sets, including Freebase and Aminer, we run them on a single Linux server with an Intel(R) Xeon E5-2673 v3 CPU@2.3GHz, 32G RAM, and one NVIDIA Ampere RTX3090-24GB GPU.

### 4.4. Results and Discussion

#### 4.4.1. Multi-Label Node Classification

To implement multi-label node classification, we refer to [28] and randomly sample 80% of the labeled nodes for training, and use the rest for testing. Based on the embedding of the training nodes, we train a logistic function to predict the most probable labels for testing nodes and compare the prediction with the ground truth labels. We report the average macro-F1 and micro-F1 scores from 10 repeated trials (Table 5). The macro-F1 calculates the global F1-score (also known as the F1-measure [45]) by counting the total true positives, false negatives, and false positives. The micro-F1 is the unweighted mean of the F1-score for each label. We highlight the best and second-place results in bold and italics, respectively.

According to the results in Table 5, CoarSAS2hvec consistently outperforms the competitors. Furthermore, CoarSAS2vec, a combination of the traditional loss function and the CoarSAS, shows competitive performances against baselines. This validates the contribution of our new sampling strategy proposed. In addition, the meta-path-based methods (metapath2vec, HeteSapceyWalk) achieve more competitive performances on Aminer, indicating that the semantic information is distinguishable on this data set. DGI demonstrates the feasibility of extracting useful network information by GNN, which is, however, limited on large data sets. For Freebase, satisfactory macro-F1 and micro-F1 are not simultaneously achieved due to its serious class imbalance.

#### 4.4.2. Community Detection

Same as the node classification task, we focus on the domain with labeled nodes in each network and regard nodes with the same label as a ground-truth community. Specifically, we cluster the labeled nodes based on their embeddings using the k-means algorithm, and evaluate the clusters using normalized mutual information (NMI), which is the normalized mutual information score [46]. NMI = 0 means two samples have no mutual information and NMI = 1 corresponds to a perfect correlation. We report the average NMI from 10 repeated trials (Table 6).

CoarSAS2hvec again outperforms other baselines and CoarSAS2vec is the second best, demonstrating the efficiency of the CoarSAS strategy. BHIN2vec works well on ACM and DBLP but fails to perform competitively on Aminer, indicating that solely balancing the network samples among node pair types is not enough for community detection. Finally, although CoarSAS2hvec surpasses all the competitors on Freebase, none of the methods learn meaningful clustering on this data set.

### 4.5. Ablation Studies

In this section, we separately explore the role of three components in CoarSAS2hvec.

#### 4.5.1. The Impact of CoarSAS

We particularly calculate the percentage of self-pairs in all node pairs (Table 7). We merge LINE, DGI, and HeGAN under the label “Edge”, which does not rely on node pair samples. In the samples of the random-walk-based methods, such as DeepWalk, metapath2vec, HeteSpaceyWalk, and BHIN2vec, the percentage of self-pairs can be nearly 20%. This would significantly reduce the number of informative samples.

To further quantify the quality of information collected by different sampling methods, we apply information entropy to characterize the “amount of information” [47,48]. For a collection of events $X=\{x_{1},\ldots ,x_{n}\}$, the information entropy H(X) is calculated as
(9)$$H\left(X\right)=-\sum_{i=1}^{n}P\left(x_{i}\right)\log\left(P\left(x_{i}\right)\right),$$
where $P(x_{i})$ is the probability that event $x_{i}$ occurs. The higher the entropy H(X) is, the more informative the collection of events *X* would be.

Compared to directly measuring the entropy of the network [49], we focus on measuring the entropy of the samples. We can apply Equation (Equation 9) to characterize the node pair samples collected, S, as
(10)$$H\left(S\right)=-\sum_{v,C_{v}\in S}\sum_{c\in C_{v}}P\left(\left(v,c\right)\right)\log P\left(\left(v,c\right)\right),$$
where P((v,c)) is the relative frequency of node pair (v,c) in S. Since self-pairs do not bring any useful information for the proximity between nodes, we remove them in the S(G). Equation (Equation 10) can be more explicitly expressed as
(11)$$\begin{array}{cc} H(S)= & -\sum_{v\in V}\sum_{c\in C_{v}}\frac{\#(v,c)}{\sum_{v\in V}\sum_{c\in C_{v}}\#\left(v,c\right)}\\ & \cdot \log\frac{\#(v,c)}{\sum_{v\in V}\sum_{c\in C_{v}}\#\left(v,c\right)},c\neq v. \end{array}$$

The information entropy H(S) of samples collected by different methods are presented in Table 7. We merge the results of edge-based methods under the label “Edge”. It is noteworthy that the performances of the methods on a data set are positively correlated with the H(S(G)) of samples collected by them to some extent. This is in line with our intuition that an algorithm can learn the system better with more information used for training and can consequently give a better performance in each task. Further studies on this issue will be investigated in future works.

The H(S(G)) of samples by CoarSAS2hvec is the highest among all methods in all data sets, confirming that CoarSAS is capable of collecting more informative samples, which further advances its performance in node classification and community discovery. This is adequately reflected by the performance of CoarSAS2vec, the method that applies the traditional skip-gram++ model on samples collected by our sampling method. With the same loss function, CoarSAS2vec outperforms metapath2vec, confirming the benefits of collecting high entropy.

#### 4.5.2. The Impact of Network Coarsening

We separately explore the contribution from the network coarsening by performing a parameter analysis on *a*, *b*, and the selected node types $T^{R}$.

The Impact of *a* and *b*.

*a* controls the ratio of each coarsening step, and *b* controls the times of the coarsening step. We depict the results of community detection on the ACM data set by different values of *a* and *b* in Figure 3a. When no coarsening is taken (a=0 or b=0), the performance is roughly the same as or slightly better than other baseline methods. The performance is improved when one round of network coarsening is taken and peaks when a=0.2 and b=4. Then, the performances decrease as *a* and *b* continue growing, which is due to the fact that the over-coarsening encourages more sampling on the original HIN, weakening the contributions of the sampling on coarsened networks.

The impact of removed node type $T^{R}$.

We depict the results of community detection on ACM, DBLP, and Aminer in Figure 3b. We omit the results on Freebase because their values are too small compared to the foregoing data sets. The reserved node type $T-T^{R}$ is shown on the top of each bar, where “N” corresponds to $T^{R}=T$. Directly coarsening the network can improve the effectiveness of the embedding results. Appropriately preserving certain node types can further enhance the gain. The reserved node type emphasizes the high-order relationships between nodes of this type, which plays a similar role to meta-path or meta-graph, but is more informative and requires less prior knowledge. The result also indicates that a high order proximity between nodes of a specific type in HIN may be more informative than direct connections.

#### 4.5.3. The Impact of Loss Function

The fact that CoarSAS2hvec outperforms CoarSAS2vec demonstrates the advance by the new loss function in Equation (Equation 8). It is then natural to ask that if the same loss function is applied in other random-walk-based methods, would the performance be improved? To answer this, we take the community detection task on the ACM data set. Equation (Equation 8) is applied in four methods from the baseline, and the results by the original method and the variant are reported in Table 8.

The performance is worse when the original loss function is replaced. Therefore, simply adopting the loss function of CoarSAS2hvec is not an appropriate choice. The new loss function is optimal only when the quality of the samples is improved. The above conclusion is also valid on the other data sets and node classification tasks; we omit them here for brevity.

### 4.6. Scalability and Convergence Analyses

We provide some additional analyses of CoarSAS2hvec for a more comprehensive understanding.

*Scalability.*Figure 4a shows the sampling costs of the SAS, coarsening, and training step of CoarSAS2hvec on each data set. The result shows that the computational complexity of SAS is roughly linear to |V| of the network. The computational complexity of coarsening and training steps are linear to |V|log|V| of the network. This conclusion is consistent with the foregoing complexity analysis. Considering Deepwalk [8] as a reference, the computational complexity of its sampling step is O(|V|) and the computational complexity of its training step is O(log|V|).

*Convergence.*Figure 4b depicts the convergence curve of the training loss, which reaches the steady-state after only a few epochs.

## 5. Conclusions

To summarize, we identify two issues in the random-walk-based HIN embedding methods that may limit the final performance. First, self-pairs may reduce the collection of informative samples. Second, hub nodes may be over-represented in sampled node pairs. To cope with the issue of imbalanced sampling, we propose a new sampling strategy: CoarSAS. The sampling method is further combined with a specifically optimized loss function, forming a new embedding method: CoarSAS2hvec.

Future directions include using the method to analyze different systems characterized by HIN, such as the scientific disciplines, the individual careers of scientists, the topic evolution in online forums, and more [50,51,52,53,54]. It is also interesting to explore whether the information entropy can be universally applied to predict the performance of an algorithm. All these questions will be addressed in later studies.

## Figures and Tables

**Figure 1 entropy-24-00276-f001:**
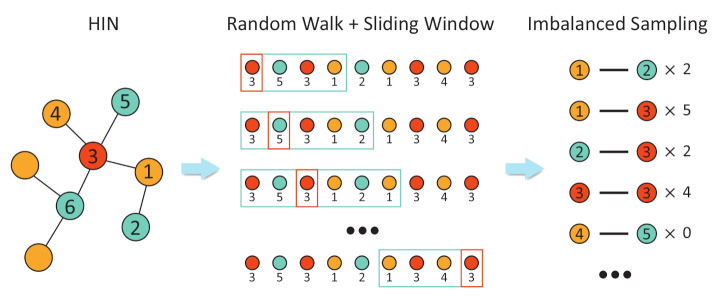
The imbalanced sampling as a result of the random walk and fixed sliding window on HIN.

**Figure 2 entropy-24-00276-f002:**
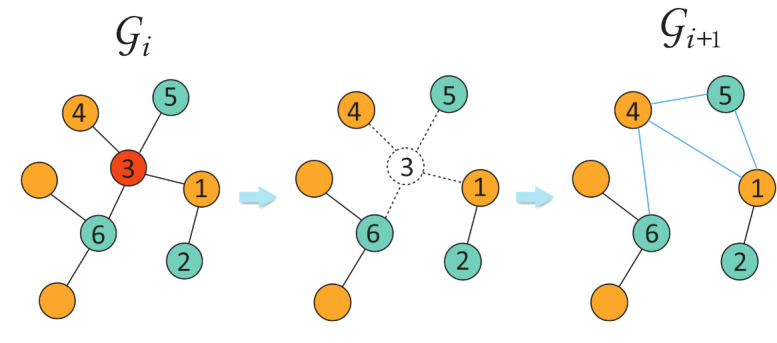
Heterogeneous network coarsening process.

**Figure 3 entropy-24-00276-f003:**
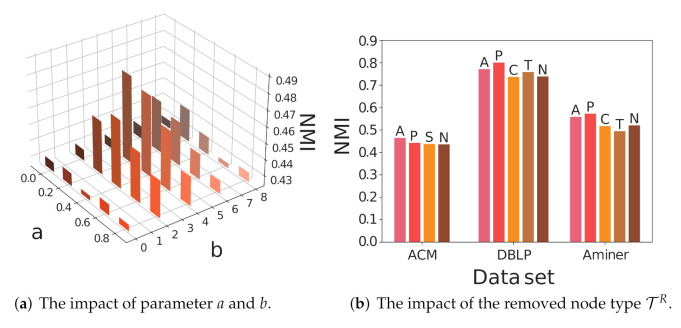
Parameter analysis of CoarSAS2hvec.

**Figure 4 entropy-24-00276-f004:**
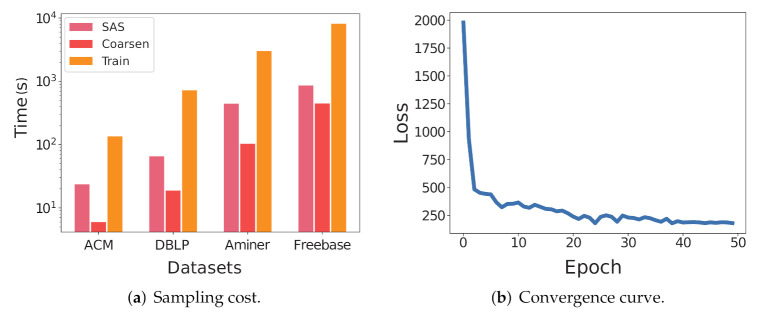
Scalability and convergency of CoarSAS2hvec.

**Table 1 entropy-24-00276-t001:** Main notations used in the paper.

Notation	Description
G	Heterogeneous information network
V	Node set of HIN
E	Edge set of HIN
T	Node type set of HIN
*v*	A node v∈V
N(v)	Neighbor nodes of *v*
$T_{N(v)}$	Neighbor node type set of *v*
S, S(G)	Sampled pairs, sampled pairs of G
$G_{i}$	Coarsened HIN after i times coarsening
*R*	The nodes to be removed in coarsening
$q_{v}$, q(G)	Sampling time of *v*, total sampling times on G
$C_{v}$	Context node set of *v* in S
$c_{k}^{\prime }$	The *k*-th negative context node $c^{\prime }\notin C_{v}$
W	The relation type indicator matrix
v	The embedding vector of *v*
c	The context embedding vector of $c\in C_{v}$

**Table 2 entropy-24-00276-t002:** Description of data sets.

Data Set	#Node	#Node	#Edge	#Edge	Target	Classes
	Types	Nums	Types	Nums		
ACM	3	11,246	2	13,407	paper	3
DBLP	4	37,791	3	41,794	author	4
Aminer	4	439,448	3	875,214	author	10
Freebase	8	164,473	36	355,072	book	7

**Table 3 entropy-24-00276-t003:** Meta-paths and meta-graphs used for each data set.

Data Set	Node Types	Meta-Paths	Meta-Graphs
ACM	A: Author	APA	[PAP || PSP]
	P: Paper	PAP	
	S: Subject	PSP	
DBLP	A: Author	APA, APCPA, APTPA	[APA || APCPA]
	P: Paper	PAP, PCP, PTP	[APA || APTPA]
	C: Conference	CPC	[APA || APCPA || APTPA]
	T: Term	TPT	[PAP || PCP]
Aminer	A: Author	APA, APCPA, APTPA	[APA || APCPA]
	P: Paper	PAP, PCP, PTP	[APA || APTPA]
	C: Conference	CPC	[APA || APCPA || APTPA]
	T: Type	TPT	[PAP || PCP]
Freebase	B: Book	BB	[BB || BOB]
	F: Film	BOB	[BB || BFB]
	L: Location	BFB	[BB || BLMB]
	M: Music	BPB	[BB || BPSB]
	O: Organization	BUB	[BB || BFB || BLMB]
	P: Person	BLMB	[BB || BFB || BPSB]
	S: Sport	BPSB	[BB || BFB || BOUB]
	U: Business	BOUB	[BB || BFB || BLMB || BPSB]

**Table 4 entropy-24-00276-t004:** Hyper-parameters of CoarSAS2hvec on different data sets.

Data Set	*a*	*b*	$T^{R}$	q(G)	*p*	*l*
ACM	0.3	3	P, S	200∗|V|	0.5	5
DBLP	0.3	3	A, C, T	200∗|V|	0.5	5
Aminer	0.2	4	A, C, T	200∗|V|	0.5	5
Freebase	0.6	2	F, L, O, P, S, U	200∗|V|	0.5	5

**Table 5 entropy-24-00276-t005:** Results of node classification; the best and second-place results are highlighted in bold and italics, respectively.

Method	ACM		DBLP		Aminer		Freebase	
	Macro-F1	Micro-F1	Macro-F1	Micro-F1	Macro-F1	Micro-F1	Macro-F1	Micro-F1
LINE	0.787	0.794	0.912	0.916	0.887	0.891	0.105	0.484
DGI	0.790	0.800	0.859	0.854	-	-	0.103	0.478
HeGAN	0.786	0.793	0.888	0.894	0.884	0.905	0.133	0.485
NSHE	0.797	0.823	0.897	0.901	0.875	0.887	0.107	0.482
Deepwalk	0.789	0.796	0.888	0.894	0.884	0.905	0.156	0.435
HIN2vec	0.432	0.762	0.904	0.910	0.862	0.875	0.118	0.475
Metapath2vec	0.720	0.729	0.907	0.914	*0.913*	*0.918*	0.134	0.436
HeteSpaceyWalk	0.750	0.759	0.905	0.911	0.888	0.893	0.154	0.442
BHIN2vec	0.750	0.759	0.905	0.911	0.890	0.893	0.153	0.416
CoarSAS2vec	*0.805*	*0.825*	*0.915*	*0.919*	0.909	0.917	*0.158*	*0.504*
CoarSAS2hvec	**0.824**	**0.842**	**0.922**	**0.929**	**0.918**	**0.926**	**0.168**	**0.511**

**Table 6 entropy-24-00276-t006:** Results of community detection (NMI); the best and second-place results are highlighted in bold and italics, respectively.

Method	ACM	DBLP	Aminer	Freebase
LINE	0.371	0.711	0.371	0.007
DGI	0.419	0.683	-	0.011
HeGAN	0.421	0.720	0.498	0.021
NSHE	0.422	0.733	0.502	0.012
DeepWalk	0.430	0.720	0.485	0.021
HIN2vec	0.434	0.673	0.475	0.014
Metapath2vec	0.417	0.785	0.525	0.010
HeteSpaceyWalk	0.433	0.772	0.475	0.024
BHIN2vec	0.432	0.771	0.460	0.008
CoarSAS2vec	*0.439*	*0.783*	*0.542*	*0.029*
CoarSAS2hvec	**0.465**	**0.805**	**0.573**	**0.034**

**Table 7 entropy-24-00276-t007:** Results of quantitative sample analysis; the best results are highlighted in bold.

Method	ACM		DBLP		Aminer		Freebase	
	$P_{\mathrm{self}-\mathrm{pairs}}$	H(S)	$P_{\mathrm{self}-\mathrm{pairs}}$	H(S)	$P_{\mathrm{self}-\mathrm{pairs}}$	H(S)	$P_{\mathrm{self}-\mathrm{pairs}}$	H(S)
Edge	-	4.241	-	5.232	-	5.942	-	6.134
HIN2vec	3.912 × 10${}^{-5}$	4.070	1.126× 10${}^{-5}$	4.601	5.560× 10${}^{-4}$	5.37	0.072	5.487
DeepWalk	0.162	5.045	0.056	6.194	0.129	6.824	0.142	6.327
Metapath2vec	0.194	4.918	0.018	6.236	0.002	6.806	0.002	5.023
HeteSpaceyWalk	0.161	5.101	0.087	6.067	0.138	6.790	0.079	6.439
BHIN2vec	0.141	5.093	0.073	6.053	0.127	6.716	0.079	6.311
CoarSAS2hvec	0	**5.396**	0	**6.704**	0	**7.193**	0	**6.621**

**Table 8 entropy-24-00276-t008:** Community detection on ACM data set (NMI).

Method	Original	2hvec
Deepwalk	0.430	0.381
Metapath2vec	0.417	0.409
HeteSpaceyWalk	0.433	0.362
BHIN2vec	0.432	0.397

## Data Availability

Publicly available datasets were analyzed in this study. ACM and DBLP can be found here: http://shichuan.org/HIN_dataset.html, accessed on 10 February 2022. Aminer can be found here: https://ericdongyx.github.io/metapath2vec/m2v.html, accessed on 10 February 2022. Freebase can be found here: https://github.com/yangji9181/HNE/tree/master/Data, accessed on 10 February 2022.

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
