# Peer review of "CoarSAS2hvec: Heterogeneous Information Network Embedding with Balanced Network Sampling"

_entropy, 2022, doi:10.3390/e24020276_

Round 1

Reviewer 1 Report

  In this paper the authors present a new method of study of heterogeneous information networks, where, by applying a different kind of sampling to the existing methods, they acquire a greater amount of information from a given network. A thorough comparison has been presented between this method and equivalent ones from the literature in both node classification and community detection. Considering the presented values and variables the CoarSAS method seems to give better results. In figure 4(b), the authors state that the steady state is reached after only a few epochs, but despite the rapid decline in the beginning of the graph the values seem to be declining until its end not appearing to having reached a minimum value. Therefore, it might be informative to the reader for the authors to include an additional number of epochs to this diagram or offer an additional one showcasing the actual converged minimum value of the loss function. Lastly the authors should fix the missing references in lines 76, 158 and 222. Overall, this paper is of good quality, presenting novel work with adequate detail and containing in depth comparisons to the other existing models. Therefore, I recommend this paper to be published with minor revisions addressing the aforementioned issues.

Author Response

Point 1: In figure 4(b), the authors state that the steady state is reached after only a few epochs, but despite the rapid decline in the beginning of the graph the values seem to be declining until its end not appearing to have reached a minimum value. Therefore, it might be informative to the reader for the authors to include an additional number of epochs to this diagram or offer an additional one showcasing the actual converged minimum value of the loss function.

Response 1: Thank you for the instructive advice. We updated the figure 4(b) by a new loss curve with 50 epochs. We believe that the current version of the figure could give a clearer picture of the final steady state.

Point 2: The authors should fix the missing references in lines 76, 158 and 222.

Response 2:  This is indeed our overlook. We have fixed these missing references in the revised manuscript. 

Reviewer 2 Report

The work considers heterogeneous information network embedding seeking to address the problem of oversampling of hub nodes in random walks leading to overrepresentation (bias) in node pair samples that negatively impact the node representation/embedding. The authors' solution is CoarSAS2hvec, which combines a self-avoid short sequence sampling plus a coarsening step to balance the overrepresentation of context nodes. The approach also uses a modified Skip-Gram with negative sampling model loss function and measure the amount of information captured by their and the baseline methods using entropy.

1) First, several minor issues that should be addressed: There are a few citation errors that need correcting. Examples: line 75 "community detection [? ]", lines 158-159 "networks[? ]", and line 222 "GAT [? ]". There are also a few places the language could be improved. Examples: line 108 "the walker successive visits" (successively visits or visits in succession), line 353 "as high as close to 20%." (nearly 20% or just shy of 20%), line 326 "Since self-pairs do bring any useful information for the proximity between nodes" (do not), line 356 "Appropriate preserving certain node types can further enhance the gain." (Appropriately), etc. A careful rereading is needed to improve the text. Also I don't think algorithm 1 is ever mentioned in the text. Perhaps it should be addressed in section 3.3. Please relabel the y-axis in Figure 4(a) as "Time (s)"; I presume it is measured in seconds.

2) Can the authors address the reasoning behind the choices of the hyperparameter settings? Was a sensitivity analysis conducted and was the optimization of the performance of a baseline method a selection consideration or only the authors' method performance? Or were the parameters selected from prior works (cite and justify/argue them)?

3) The authors discuss scalability and convergence in section 4.6. Did the authors compare the time complexity/sampling cost of their approach versus the alternative baseline approaches? This would help determine what the cost is (if any) for obtaining the improved performance given by the authors CoarSAS2hvec approach.

Author Response

Point 1:  First, several minor issues that should be addressed: There are a few citation errors that need correcting. Examples: line 75 "community detection [? ]", lines 158-159 "networks[? ]", and line 222 "GAT [? ]". There are also a few places the language could be improved. Examples: line 108 "the walker successive visits" (successively visits or visits in succession), line 353 "as high as close to 20%." (nearly 20% or just shy of 20%), line 326 "Since self-pairs do bring any useful information for the proximity between nodes" (do not), line 356 "Appropriate preserving certain node types can further enhance the gain." (Appropriately), etc. A careful rereading is needed to improve the text. Also I don't think algorithm 1 is ever mentioned in the text. Perhaps it should be addressed in section 3.3. Please relabel the y-axis in Figure 4(a) as "Time (s)"; I presume it is measured in seconds.

Response 1: We wish to thank the Referee for pointing out our overlook. We have fixed the missing references in lines 76, 158 and 223. We have carefully reread and fixed the inaccuracies and typos in line 75, 108, 220, 252, 304, 329, 332, 362 as well. To address the omission of the algrithm1, we have added a reference in line 192 at the end of section 3.2 and slightly modified the beginning of section 3.3 for a better articulation. We also have relabeled the y-axis in Figure 4(a) as "Time (s)".

Point 2:  Can the authors address the reasoning behind the choices of the hyperparameter settings? Was a sensitivity analysis conducted and was the optimization of the performance of a baseline method a selection consideration or only the authors' method performance? Or were the parameters selected from prior works (cite and justify/argue them)?

Response 2: The parameters are selected from prior works. We set the embedding dimension d = 128 following the most related works [1-3] because this is a common setting among them and set the rest of hyper-parameters following [4] because it provides a uniform parameter setting for methods of different categories. The references and the explanations are added in line 261-268 at section 4.3.1.

[1] Dong, Y.; Chawla, N.V.; Swami, A. Metapath2vec: Scalable Representation Learning for Heterogeneous Networks. SIGKDD. ACM, 2017, p. 135–144.

[2] Lee, S.; Park, C.; Yu, H. BHIN2vec: Balancing the Type of Relation in Heterogeneous Information Network. CIKM. ACM, 2019, p.619–628.

[3] Jiang, J.Y.; Li, Z.; Ju, C.J.T.; Wang, W. MARU: Meta-Context Aware Random Walks for Heterogeneous Network Representation Learning. CIKM. ACM, 2020, p. 575–584.

[4] Cen, Y.; Zou, X.; Zhang, J.; Yang, H.; Zhou, J.; Tang, J. Representation Learning for Attributed Multiplex Heterogeneous Network. SIGKDD. ACM, 2019, p. 1358–1368.

Point 3: The authors discuss scalability and convergence in section 4.6. Did the authors compare the time complexity/sampling cost of their approach versus the alternative baseline approaches? This would help determine what the cost is (if any) for obtaining the improved performance given by the authors CoarSAS2hvec approach.

Response 3: We agree with the Referee that it is valuable to compare the complexity of our approach with the baselines. Since our method is based on random walk and Skip-gram kernel, the complexity is close to that of Deepwalk. In the revised manuscript, we added the discussion about complexity in line 387-389.